# Anti-Inflammatory and Antimicrobial Activities of Portuguese *Prunus avium* L. (Sweet Cherry) By-Products Extracts

**DOI:** 10.3390/nu14214576

**Published:** 2022-10-31

**Authors:** Ana R. Nunes, José D. Flores-Félix, Ana C. Gonçalves, Amílcar Falcão, Gilberto Alves, Luís R. Silva

**Affiliations:** 1CICS-UBI—Health Sciences Research Centre, University of Beira Interior, Av. Infante D. Henrique, 6200-506 Covilhã, Portugal; 2CNC—Centre for Neuroscience and Cell Biology, Faculty of Medicine, University of Coimbra, 3004-504 Coimbra, Portugal; 3Laboratory of Pharmacology, Faculty of Pharmacy, University of Coimbra, 3000-548 Coimbra, Portugal; 4CIBIT—Coimbra Institute for Biomedical Imaging and Translational Research, University of Coimbra, 3000-548 Coimbra, Portugal; 5CPIRN-UDI-IPG—Research Unit for Inland Development, Center for Potential and Innovation of Natural Resources, Polytechnic Institute of Guarda, 6300-554 Guarda, Portugal

**Keywords:** *Prunus avium* L., by-products, anti-inflammatory activity, antibacterial activity

## Abstract

The bioactivity of natural by-products in food and pharmaceutical applications is the subject of numerous studies. Cherry production and processing generates large amounts of biowaste, most of which is not used. The recovery of these by-products is essential for promoting the circular economy and to improving sustainability in the food industry. In this work, we explored the anti-inflammatory and antimicrobial potential of two different extracts from stems, leaves, and flowers of Portuguese cherries. The anti-inflammatory potential was studied on lipopolysaccharide (LPS)-stimulated mouse macrophages (RAW 264.7) by evaluating the effect of by-products on cellular viability and nitric oxide (NO) production. Disc diffusion and minimum inhibitory concentration (MIC) were used to determine antimicrobial activity. The cherry by-products had no cytotoxic effect on RAW 264.7 cells, and were able to inhibit nitrite production in a dose-dependent manner. Moreover, all aqueous infusions showed good antioxidant activity against NO radicals. Moreover, leaf extracts showed the best activity against most of the strains studied. The results revealed, for the first time, interesting anti-inflammatory and antimicrobial properties of cherry by-products. This could potentially be of interest for their therapeutic use in the treatment of inflammation-related diseases or in controlling the growth of microorganisms.

## 1. Introduction

Since ancient times, plants formed the basis for the development of many therapeutic products used in traditional medicine with significant pharmacological properties [1]. Natural products consist of various bioactive compounds that have been shown to have health-promoting properties, keep the body healthy, and protect against a variety of diseases, many of which are related to oxidative stress [2]. Nowadays, consumers are looking for healthier foods/products for their daily life. In this context, food and pharmaceutical industries are increasingly interested in finding compounds with good biological properties from natural sources [3,4]. Detailed knowledge of phytochemical composition, biological properties, safety profile, and environmental toxicity is essential for the characterization of herbal products.

*Prunus avium* L., commonly known as sweet cherry, is a high-value fruit of great nutritional and economic importance, widely distributed throughout the world [5,6,7]. The literature dealing with this specie has shown that the cherries possess interesting biological properties, and are indispensable in human nutrition [8,9,10,11]. According to various studies, the presence of phytochemicals such as phenolic compounds (flavonoids and non-flavonoids), minerals, and organic volatiles is the basis for these properties [7,12].

Cherry harvesting and industrial processing (e.g., into candies, jellies, juices, and fresh-cut fruits), produces a large amount of by-products, such as leaves, stems, seeds, and pulp. In addition, cherry flowers decompose in the soil after spring, without being used or recycled in any way [7]. Although the by-products are treated as biowaste, recent studies have shown that these basic plant materials are an excellent source of bioactive compounds that can be used in value-added products [13,14,15]. Traditionally, cherry stems in aqueous infusion have been used as diuretics, sedatives, anti-inflammatory agents, and to promote cardiovascular health [7]. Previous works developed by our research team have described the antioxidant, antiproliferative, and antidiabetic effects of *P. avium* fruits and their by-products [8,11,13,14]. However, little is known about the anti-inflammatory and antimicrobial potential of cherry by-products.

The anti-inflammatory process is a normal response of the body to an infection or injury. Immunological biochemical reactions lead to the release of reactive oxygen species (ROS) and reactive nitrogen species (RNS) and, consequently, to an increase in oxidative stress [16]. When inflammation becomes chronic or unregulated it can lead to serious pathological conditions. Therefore, the search for new alternatives that act on inflammation became the subject of the study [17]. The anti-inflammatory effect of extracts from sweet cherries has been recently studied and showed promising results, mainly due to their richness in phenolic compounds [11].

Despite the already demonstrated biological properties of *P. avium* fruits and their by-products, their antimicrobial properties are still poorly studied. A study conducted by Afonso and co-workers [10] showed that extracts from cherry stems are able to inhibit Gram-positive bacteria, probably due to the presence of phenolic acids in this by-product. Similarly, previous studies have shown that cherry stems, leaves, and flowers have a high content of these phenolic compounds [10]. In addition, the increase multidrug-resistant bacteria associated with the decrease of molecules with antibacterial properties makes it necessary to find new compounds containing molecules with this activity [18]. In this context, secondary metabolites produced by plants, which act as defense compounds against pathogens, could be a good strategy to explore.

The present work aimed to investigate the anti-inflammatory and antimicrobial properties of leaves, stems and flowers of a Portuguese *P. avium* (cv. *Saco*) in vitro. For this purpose, the effect of these by-products was analyzed in a mouse macrophage cell line (RAW 264.7) stimulated with lipopolysaccharide (LPS), as an inflammatory model. Nitric oxide (NO) production and scavenging capacity were studied. In addition, antimicrobial activity was determined by disk diffusion and minimum inhibitory concentrations (MICs) using the broth microdilution method.

## 2. Materials and Methods

### 2.1. Chemicals and Reagents

N-(1-naphthyl)ethylenediamine dihydrochloride, sulfanilamide and sodium nitroprusside dihydrate (SNP) were acquired from Alfa Aesar (Karlsruhe, Germany). Dulbecco’s modified Eagle’s medium (DMEM), fetal bovine serum (FBS), penicillin, streptomycin, trypsin-ethylenediaminetetraacetic acid solution, 3-(4,5-dimethylthiazol-2-yl)-2,5-diphenyltetrazolium bromide (MTT), dimethyl sulfoxide (DMSO), and β-nicotinamide adenine dinucleotide (NADH) were from Sigma-Aldrich (St. Louis, MO, USA). Purified water was acquired using the Milli-Qplus185 system (Millipore, Billerica, MA, USA). Müeller-Hinton agar (MHA) was purchased in VWR (Prolabo Chemicals, Radnor, PA, USA).

### 2.2. Plant Material

*P. avium* by-products, namely leaves, stems, and flowers were collected between April and June 2018 in the Fundão region (Portugal), as previously described [13]. Samples were frozen, freeze-dried in a freeze dryer lab equipment (ScanVac CoolSafe, LaboGene APS, Allerød, Denmark), reduced to a dried powder, and stored at −20 °C until analysis [13].

### 2.3. Extracts Preparation

Hydroethanolic extracts and aqueous infusions of *P. avium* by-products were prepared according to the method previously described [13]. Firstly, leaves, stems, and flowers were reduced in a dried powder. For the hydroethanolic extracts, 1 g of the dried powder of each sample was dissolved in ethanol/water (50:50, *v*/*v*), and posteriorly sonicated for 30 min. After that, the hydroethanolic solutions were shaken at room temperature for 2 h and then again sonicated for 30 min. Finally, the extracts were filtered using a 0.45 µm membrane (Millipore, Bedford, MA, USA) in a vacuum system, evaporated under reduced pressure, freeze-dried, and stored at −20 °C until further use.

Regarding to aqueous infusions, 1 g of dried powder of each sample was subjected to an infusion (100 mL of water) at 100 °C for 3 min, according to the manufacturer’s instructions for daily herbal infusions. Then, infusions were filtered as described above to hydroethanolic extracts, freeze-dried, and stored at −20 °C until further analysis. The extraction yields were reported in a previous work [13].

### 2.4. Cell Culture

RAW 264.7 cell line, was cultured in a DMEM high glucose medium, supplemented with 10% (*v*/*v*) of non-inactivated FBS, 1% penicillin/streptomycin, and 1.5 g/L sodium bicarbonate, at 37 °C in a humidified atmosphere of 95% air and 5% CO_2_. The cells were passaged using a cell scrapper, before reaching confluence and according to ATCC recommendations.

### 2.5. Anti-Inflammatory Potential

#### 2.5.1. Effect of Cherry By-Products Extracts on Cellular Viability

To determine the effects of hydroethanolic extracts and aqueous infusions of *P. avium* by-products on cellular viability, the 3-(4,5-dimethyl-thiazol-2-yl)-2,5-diphenyl tetrazolium bromide (MTT) assay was performed, as previously described [19]. Briefly, 2.5 × 10^4^ cells per well were seeded in a 96-well plate and incubated with different concentrations (50, 100, 200, 400, and 800 µg/mL) of the extracts, for 24 h, according to the described method, with minor adjustments [19]. Then, the medium was then removed and the MTT solution was added to each well (1 mg mL^−1^) and the plates were incubated, in the dark at 37 °C for 4 h. The absorbance of formazan crystals was measured at 570 nm using an XmarckTM microplate absorbance spectrophotometer (Bio-Rad, Hercules, CA, USA). Six independent experiments were performed in triplicate.

#### 2.5.2. Nitric Oxide (NO) Radicals Production

For nitrite accumulation in culture medium, cells were seeded at a density of 1 × 10^5^ per well in a 96-well plate for 24 h, according to the method described by Oliveira and co-workers [20]. After that, the medium was removed, and the cells were incubated for 2 h with different dilutions, as described in the previous section. Subsequently, 1 µg mL^−1^ LPS was used to stimulate the cells, for 22 h. Production of NO radicals was determined by measuring nitrite accumulation in the supernatant, as previously described [20]. The Griess reagent (1% (*w*/*v*) sulfanilamide in 5% (*w*/*v*) phosphoric acid and 0.1% (*w*/*v*) N-(1-naphthyl)-ethylenediamine dihydrochloride) was mixed into the supernatants, and incubated at room temperature for 30 min. Absorbance was measured at 550 nm.

#### 2.5.3. Determination of Radicals NO Scavenging Activity

NO radicals scavenging activity was evaluated according to the method explained by Jesus and co-workers [21]. Briefly, the same concentrations used in the cells were dissolved in potassium phosphate buffer (100 mM, pH 7.4) and mixed with SNP (20 mM), as described by Gonçalves and co-workers [11]. Plates were then incubated for 1 h at room temperature, under light. Griess reagent, composed by 1% sulfanilamide and 0.1% naphthylethylenediamine in 2% phosphoric acid (H_3_PO_4_), was added to the wells, and the multiwell plates were incubated for 10 min, in the dark. Then, the absorbance was measured at 560 nm. The radical scavenging activity NO was determined by comparing the absorbance values between the extracts and the control and corresponded to the mean ± standard deviations of three independent experiments, performed in triplicate.

### 2.6. Antimicrobial Potential

#### 2.6.1. Bacterial Strains and Growth Conditions

Thirteen bacterial strains were used in this work acquired from American Type Culture Collection (ATCC, Manassas, VT, USA), BCCM/LMG Bacteria Collection (Belgian Co-Ordinated Collections of Micro-organisms, Gent, Belgium), and the Spanish Type Culture Collection (Valencia, Spain). Six Gram-positive bacteria (*Micrococcus luteus* (CECT 243), *Enterococcus faecalis* (ATCC 29212), *Bacillus cereus* (ATCC 11778), *Listeria monocytogenes* LMG 16779, *Staphylococcus aureus* ATCC 25923, and *Bacillus subtilis* (CECT 35)), and seven Gram-negative bacteria (*Salmonella typhimurium* (ATCC 13311), *Pseudomonas aeruginosa* ATCC 27853, *Escherichia coli* ATCC 25922, *Klebsiella pneumoniae* (ATCC 13883), *Proteus mirabilis* (CECT 170)*, Serratia marcescens* (CECT 159) and *Acinetobacter baumannii* (LMG 1025)).

#### 2.6.2. Disk Diffusion Method

The disk diffusion test was performed according to CLSI M02-A12 (2015) for bacteria [22]. Sterile Petri dishes (90 mm Ø) were used using 25 mL of medium. A 0.5 McFarland suspensions were made from bacterial cultures in sterile saline solution. Antimicrobial disks (Filtar Lab, 6 mm Ø) were impregnated with 10 µL each of a hydroethanolic extract or an aqueous infusion of *P. avium* leaves, stems, and flowers, following to the method previously described [23], with small modifications. Gentamycin was used as a positive control. MHA and with tryptic soy agar (TSA) medium were used for this experiment. The plates were incubated at 37 °C for 24 h. Posteriorly, the inhibition zone was visualized and measured in millimeters. The obtained results are reported as means ± standard deviations of three independent experiments.

#### 2.6.3. Evaluation of the Minimum Inhibitory Concentration (MIC)

The susceptibility of bacteria to the hydroethanolic extracts and aqueous infusions of cherry leaves, stems, and flowers was determined using the broth microdilution method, as previously described [24]. Briefly, inoculums were prepared by suspension in NaCl 0.85% (*w*/*v*), and turbidity was adjusted to 0.5 McFarland to get a final concentration of approximately 5 × 106 CFU mL^−1^. Experiments were performed in 96-well plate with a range of concentrations between 2 and 0.015 mg mL^−1^, posteriorly incubated at 37 °C for 24 h. Then, 30 µL of a resazurin solution (0.01%) was added to each well and incubated for 2 h at 37 °C. The assay was performed in triplicate, and results were reported as modal values.

### 2.7. Evaluation of P. avium by-Products Biocompatibility

The cytotoxicity of hydroethanolic extracts and aqueous infusions of sweet cherry leaves, stems, and flowers was investigated using the normal human dermal fibroblasts (NHDF) cell line. This cell line was obtained from the American Type Culture Collection (ATCC; Manassas, VA, USA) and grown in high-glucose DMEM medium supplemented with 10% FBS and 1% antibiotic/antimycotic and 25 µg mL^−1^ amphotericin B (Sigma-Aldrich, Inc., St. Louis, MO, USA). 1 × 10^4^ cells per well were seeded in 96-well plate and incubated at 37 °C in a humidified atmosphere of 95% air and 5% CO_2_ for 24 h. After, the cells were incubated with different concentrations of extracts of *P. avium* by-products (50, 100, 200, 400, and 800 µg mL^−1^), according to the method described previously [13]. After 24 h, the MTT assay described in Section 2.5.1 was performed. The absorbance was measured at 570 nm using an XmarckTM Microplate Absorbance Spectrophotometer (Bio-Rad, Hercules, CA, USA). Six independent experiments were performed in triplicate.

### 2.8. Statistical Analysis

The results were analyzed using GraphPad Prism 8 (GraphPad Software, San Diego, CA, USA). Data were expressed as mean ± standard deviations or mean ± standard error of the mean. One-way analysis variance (ANOVA) with Tukey’s multiple comparison test or Student’s *t*-test were performed to compare the effect of each concentration of *P. avium* by-products extract with controls. A *p* value < 0.05 was considered statistically significant.

## 3. Results and Discussion

### 3.1. Anti-Inflammatory Potential

The anti-inflammatory process is the body’s natural mechanism in response to infection or injury. However, if the inflammatory state is overestimated, it can contribute to the onset and development of numerous diseases, such as cancer, cardiovascular and neurological diseases [25]. Plant extracts consist of various phytochemicals, mainly phenolic compounds [6,7]. According to the literature, these compounds possess a plethora of health benefits, including anti-inflammatory properties [11,20].

Macrophages play a crucial function in the innate immune response, so in this work, we used RAW 264.7 cells as an in vitro inflammatory model to investigate the possible anti-inflammatory activity of *P. avium* stems, leaves, and flowers.

#### 3.1.1. Effect of *P. avium* By-Products on the Viability of the Raw 264.7 Cell Line

To determine whether the hydroethanolic extracts and aqueous infusions of sweet cherry leaves, stems, and flowers affect macrophages viability of, the MTT assay was performed after 24 h of exposure. The concentration of the extracts was adjusted to 50, 100, 200, 400 and 800 μg mL^−1^. As shown in Figure 1, measurement of cell viability after treatment with all tested concentrations of the two extracts from *P. avium* stems and flowers aqueous infusion resulted in cell viability ≥90% (Figure 1C,D,F). The hydroethanolic extract and aqueous infusion both from leaves showed a cell viability ≥90% in a concentrations range of 50 to 400 μg mL^−1^ and 50 to 200 μg mL^−1^, respectively (Figure 1A,B). The hydroethanolic extract of flowers showed cell viability between 80% and 90% (Figure 1E). Similar results were obtained by Orabona and co-workers [26] for *Crocus sativus* L. by-products. Based on these results, we can assume that the concentrations used are safe and non-toxic for further studies.

The biological properties of secondary plant metabolites have been extensively studied [7,27,28]. According to previous studies about phytochemical composition of *P. avium* by-products, it is known that cherry leaves, stems, and flowers are composed by several phenolic compounds, namely hydroxycinnamic acid, flavonols, flavan-3-ols, and flavanones [13,21]. Among several phenolics present in these by-products, the 3-caffeoylquinic acid cis, 5-caffeoylquinic acid trans, quercetin 3-*O*-rutinoside, quercetin 3-*O*-glucoside, 3-coumaroyl-5-caffeoylquinic acid, and 3-coumaroyl-4-caffeoylquinic acid are the most abundant [13]. The maintenance of RAW 264.7 cells viability when exposed to several concentrations and different extracts may be related to the presence of these hydroxycinnamic acids and flavonoids. Furthermore, these phenolic compounds are described as being able to reduce inflammation [29]. In addition, they are molecules with low cytotoxicity [29,30].

#### 3.1.2. Effect of *P. avium* By-Products on NO Production

Numerous inflammatory mediators are released as part of the inflammatory process. In addition, macrophages produce NO in large quantities, and it is known that natural products have shown a good response to this increase [31]. Lipopolysaccharide are extracellular elements of Gram-negative bacteria and possess potent stimuli for monocytes and macrophages. When these cells are activated by a stimulus induced by LPS, they produce inflammatory mediators, including NO, and release them via the regulation of pro-inflammatory factors [32]. Therefore, in this work, we investigated the anti-inflammatory potential of hydroethanolic extracts and aqueous infusions of cherry leaves, stems, and flowers through their ability to inhibit the production of NO in stimulated RAW 264.7 macrophages. An inflammatory stimulus was elicited by LPS (1 μg mL^−1^), to simulate an inflammatory state. The production of nitrites (NO stable metabolites), was measured after cell exposure to *P. avium* by-products extracts by the colorimetric Griess reaction.

As shown in Figure 2, all extracts of cherry by-products were able to decrease the production of NO, in LPS-stimulated RAW 264.7 cells, in a dose-dependent manner. In particular, the hydroethanolic extracts of leaves and stems and the aqueous infusions of flowers were able to decrease NO to below 50% at extract concentrations of 400 μg mL^−1^ or more (Figure 2A,C–F). Interestingly, both extracts of flowers were promising, and showed a very significant reduction in NO, at extract concentrations of 200 μg mL^−1^ or greater (Figure 2E,F). To our knowledge, no similar studies have been performed with sweet cherry by-products. However, similar profiles were obtained for sweet cherry phenolic-rich fractions of the same variety of by-products [11].

The anti-inflammatory effects of flavonoids are well described in the literature and have been used for a long time [33]. These compounds are exogenous antioxidants capable to reducing ROS by inhibition of specific enzymes (e.g., NO and xanthine oxide synthases), and also by regulation of ion channels. Moreover, flavonoids reduce the inflammatory state by inhibiting regulatory substances, such as NF-κB, activator protein-1 (AP-1), interleukin-1 beta (IL-1β), tumor necrosis factor-alpha (TNF-α), interleukins 6 and 8, and COX-2 [33]. Recently, a study conducted by Gonçalves and co-workers [11], demonstrated that colored and non-colored fractions of *P. avium* fruits decreased the expression of COX-2, probably due to the abundance of anthocyanins [34]. Several flavonoids were found in *P. avium* leaves, stems, and flowers, including non-flavonoid compounds such as phenolic acids [13,15]. Caffeoylquinic acid, *p*-coumaric acid, kaempferol derivatives, and quercetin 3-*O*-rutinoside are phenolics present in extracts of cherry by-products and they have been proven to decrease NO production on LPS-stimulated RAW 264.7 cells [11,35,36,37].

#### 3.1.3. Effect of *P. avium* By-Products on NO Scavenging Activity

To determine whether the decrease in production of NO was due to the scavenging activity of extracts from *P. avium* leaves, stems, and flowers, the NO scavenging activity was considered. As shown in Figure 3, all hydroethanolic extracts and aqueous infusions of cherry by-products showed significant scavenging activity in a concentration-dependent manner. The aqueous infusions of cherry stems, flowers, and leaves were the most active (Figure 3B,D,F) showing a scavenging activity greater than 50% (IC_50_ values of 111.4 ± 0.31, 149.3 ± 0.94, and 183.5 ± 0.72 μg mL^−1^, respectively). Hydroethanolic extracts and aqueous infusions of leaves, stems, and flowers demonstrated an excellent scavenging activity against NO radicals compared to ascorbic acid (IC_50_ = 767.67 ± 4.04 μg mL^−1^) (data not shown). The results obtained are in agreement with previous studies [21].

The phenolic compounds are natural molecules with a relevant antioxidant potential. Oxidative stress can contribute to the inflammatory process through the activation of several anti-inflammatory mediators. In this context, phenolics appear as quite useful compounds in the inhibition of oxidative stress. The antioxidant properties of *P. avium* by-products previously studied [13,21] are associated with the capacity to capture free radicals. The chemical structure of phenolic compounds in sweet cherry leaves, stems, and flowers is capable of transferring hydrogen atoms to radicals, thus reducing their concentration [11]. In addition, the NO scavenging activity of the extracts may help to break the chain of reactions triggered by excessive formation of NO and, thus, prevent the development of diseases, which also favors their use in the treatment of inflammatory disorders.

### 3.2. Antimicrobial Potential

Screening of antimicrobial activity of hydroethanolic extracts and aqueous infusions of *P. avium* leaves, stems, and flowers was evaluated using several Gram-positive and Gram-negative bacteria. For this purpose, the disk-diffusion method was first performed, which measures the influence of the *P. avium* extracts on the normal growth of bacteria on agar plates. The larger the diameter without growing bacteria around the paper disc impregnated with the extract, the higher the growth inhibitory properties of the extract. The diameters of the inhibition zones are shown in Table 1. Thus, higher antimicrobial activity against *B. cereus* (ATCC 11778) was observed with the hydroethanolic extract of stems, followed by stems aqueous infusion (inhibition halo of 7.24 ± 0.14 mm and 4.06 ± 0.2 mm, respectively) (Table 1). In addition, the hydroethanolic extract of stems was also able to inhibit the growth of *S. aureus* (ATCC 25923), with an inhibition halo of 3.04 ± 0.16 mm. According to the study conducted by Afonso and co-workers [10], extracts from cherry stems were also found to be more effective, for Gram-positive bacteria. The strain *P. mirabilis* (CECT 17) was the most resistant to the hydroethanolic extract from leaves, with an inhibition halo of 0.32 ± 0.15 mm. In general, the hydroethanolic extracts of *P. avium* by-products were those that showed the highest microbial activity against Gram-positive bacteria. These by-products showed almost no antimicrobial activity against Gram-negative bacteria (Table 1).

The antimicrobial activity of *P. avium* by-products has been associated with the content of phenolic compounds. Indeed, several authors reported that phenolics, such as *p*-coumaric acid, *p*-coumaroylquinic acid derivative, sakuranetin, neochlorogenic and chlorogenic acids, and catechin are some compounds involved in the antimicrobial properties of plant extracts [11,38,39,40]. Aromatic hydroxyl groups present in many phenolics have high affinity for bacterial membranes, interfering with membrane and cytoplasmic organelles, which may lead to bacterial death [10,41,42].

According to visual MIC, high activity of the hydroethanolic extract of leaves followed by the aqueous infusion leaves was observed, with MIC values ranging from 0.016 to 2 mg mL^−1^ for most of the bacterial strains studied (Table 2). The obtained results show that the by-products of *P. avium* have inhibitory activity against more than half of the tested strains, namely, *S. tiphymurium*, *E. faecalis*, *S. marcescens*, and *B. cereus* (Table 2). It can be concluded that the chemical composition of sweet cherry leaves, stems, and flowers contains a wide range of phytochemicals with antimicrobial properties. Many mechanisms may lead to stronger antimicrobial activity, such as the chemical composition of the by-products and the method of extraction of the bioactive compounds. According to Gullon and collaborators [43], the antimicrobial activity can be associated with the combination of several bioactive compounds.

### 3.3. Effect of P. avium By-Products on NHDF Cell Line

To understand the biocompatibility of *P. avium* by-product extracts with human cells, the effects of hydroethanolic extracts and aqueous infusions of leaves, stems, and flowers were studied in NHDF cells (Figure 4). The results show that, both extracts from stems increased significantly the cellular viability of these cells (Figure 4C,D). Although aqueous infusions of leaves and flowers slightly reduced the viability of NDHD, and considering ISO 1093:5-2009, it is possible that the concentrations used did not present cytotoxicity when compared to untreated cells (Control) (Figure 4B,F).

The biological properties of the extracts are related to their composition of phenolic compounds. In the literature, several studies have demonstrated that these phytochemicals, isolated or in synergy, contribute to the elimination of ROS and, consequently to the reduction of oxidative stress, promote the maintenance of healthy skin through collagen production [44,45,46,47]. Detailed knowledge of the chemical composition and biological properties of *P. avium* by-products in conjugation with its biosafety profile, biocompatibility, and use is essential for biomedical applications [48].

## 4. Conclusions

This work is the first to report the anti-inflammatory and antimicrobial activity of leaves, stems, and flowers of *P. avium* from the Fundão region (Portugal). Both sweet cherry by-products extracts were found to be safe and non-cytotoxic to RAW 264.7 and NDHF cell lines. The hydroethanolic extracts of leaves and stems and the aqueous infusion of flowers showed promising results suppressing inflammation by reducing the production of NO by LPS-activated RAW 264.7 macrophages. In addition, all aqueous infusions of the by-products, especially cherry flowers, showed excellent scavenging activity against NO radicals. In addition, cherry leaves, stems, and flowers were found to be more effective against Gram-positive bacteria. Moreover, the hydroethanolic extract of the leaves and the aqueous infusion of the leaves showed great antimicrobial capacity for most of the tested bacteria. Our results demonstrate, for the first time, that *P. avium* by-products are a good source of phenolics with relevant anti-inflammatory and antimicrobial potential, highlighting the importance of using these biowastes as bioactive compounds for therapeutic use in inflammation and control the microbial growth.

## Figures and Tables

**Figure 1 nutrients-14-04576-f001:**
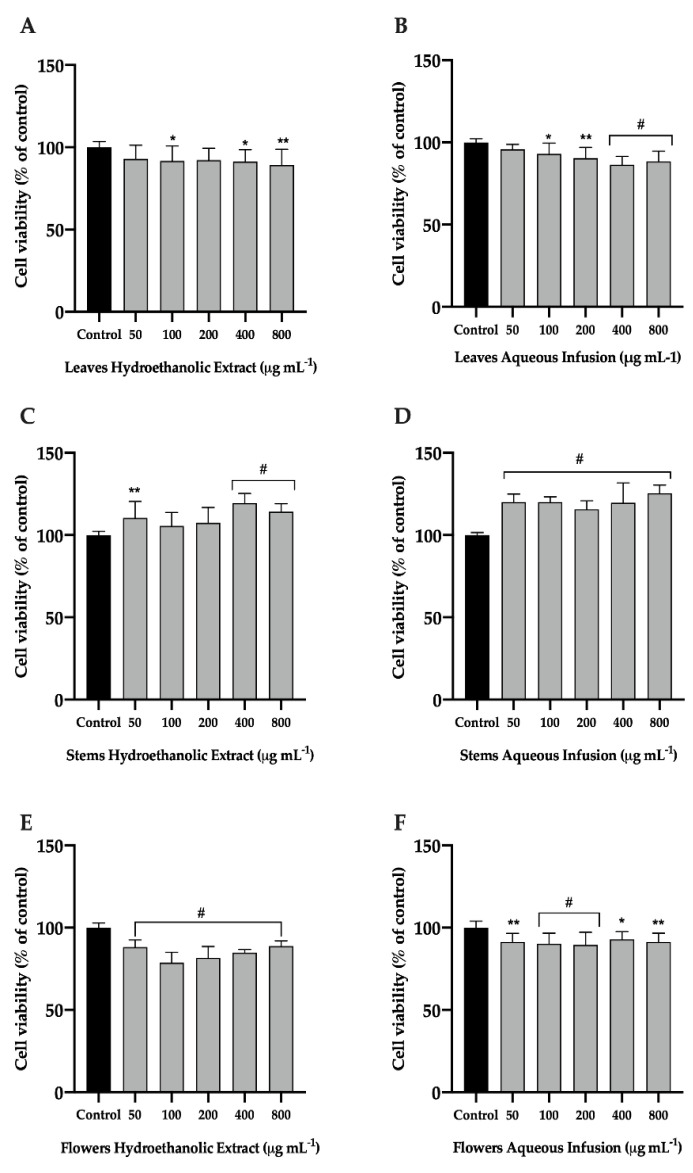
Effect of hydroethanolic extract (**A**,**C**,**E**) and aqueous infusion (**B**,**D**,**F**) of *Prunus avium* L. leaves, stems, and flowers on RAW 264.7 macrophages. Cells were plated and submitted to increasing concentrations of extracts for 24 h. 3-(4,5-dimethylthiazol-2-yl)-2,5-diphenyltetrazolium bromide (MTT) assay was performed to assess cell viability. The presented data correspond to the means ± standard deviation of three independent assays and are represented as percentage (%) of control cells. Statistical analysis: One-way ANOVA was performed for each concentration compared to control; * *p* value < 0.05, ** *p* < 0.01, and # *p* < 0.001 were considered significant.

**Figure 2 nutrients-14-04576-f002:**
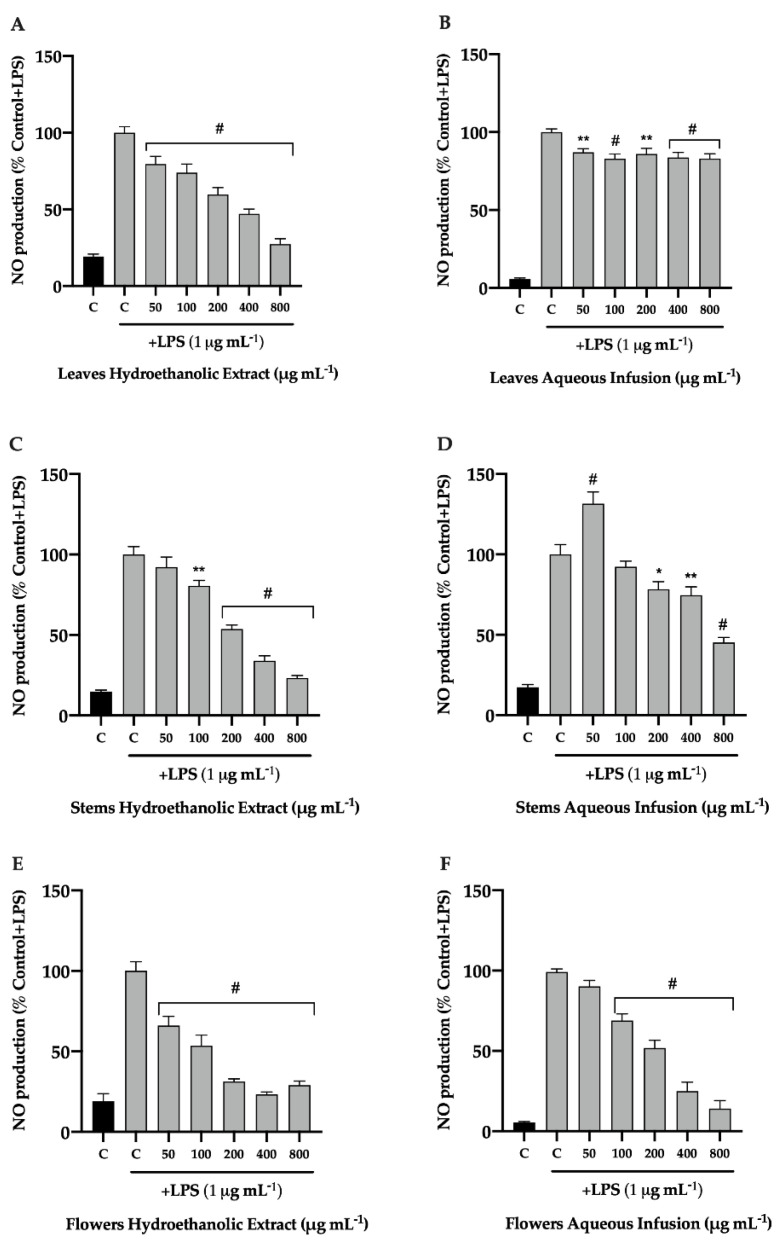
Effect of hydroethanolic extract (**A**,**C**,**E**) and aqueous infusion (**B**,**D**,**F**) of *Prunus avium* L. leaves, stems, and flowers on macrophage nitric oxide (NO) production upon an inflammatory stimulus. Cells were plated and exposed to Ctrl medium exposed to increasing concentrations of extracts for 24 h, in the presence of lipopolysaccharide (LPS) (1 μg mL^−1^). Griess assay was performed to assess nitrites levels in the supernatant. The presented data correspond to the means ± standard error of the mean of three independent assays and are represented as percentage (%) of control cells (C) cells exposed to LPS (Ctrl LPS). Statistical analysis: One-way ANOVA was performed for each concentration compared to control with LPS; * *p* value < 0.05, ** *p* < 0.01, and # *p* < 0.001 were considered significant.

**Figure 3 nutrients-14-04576-f003:**
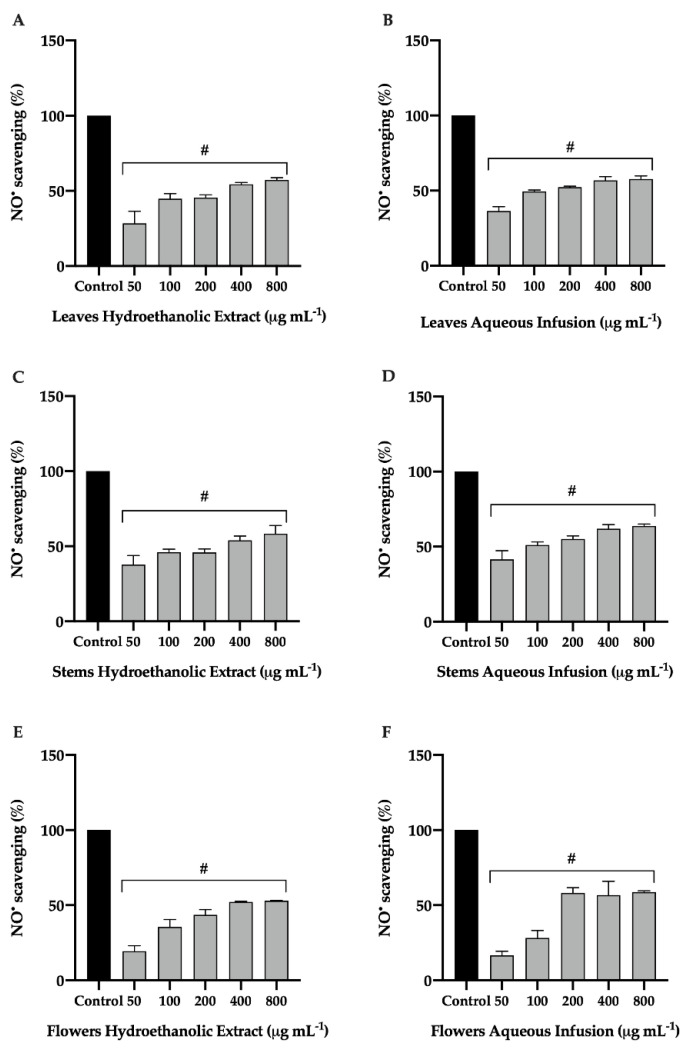
Effect of hydroethanolic extract (**A**,**C**,**E**) and aqueous infusion (**B**,**D**,**F**) of *Prunus avium* L. leaves, stems, and flowers on NO scavenging activity. Griess assay was performed to assess NO scavenging in supernatant. The presented data correspond to the means ± standard deviation of three independent assays and are represented as percentage (%) of control. Statistical analysis: One-way ANOVA was performed for each concentration compared to control; # *p* < 0.001 was considered significant.

**Figure 4 nutrients-14-04576-f004:**
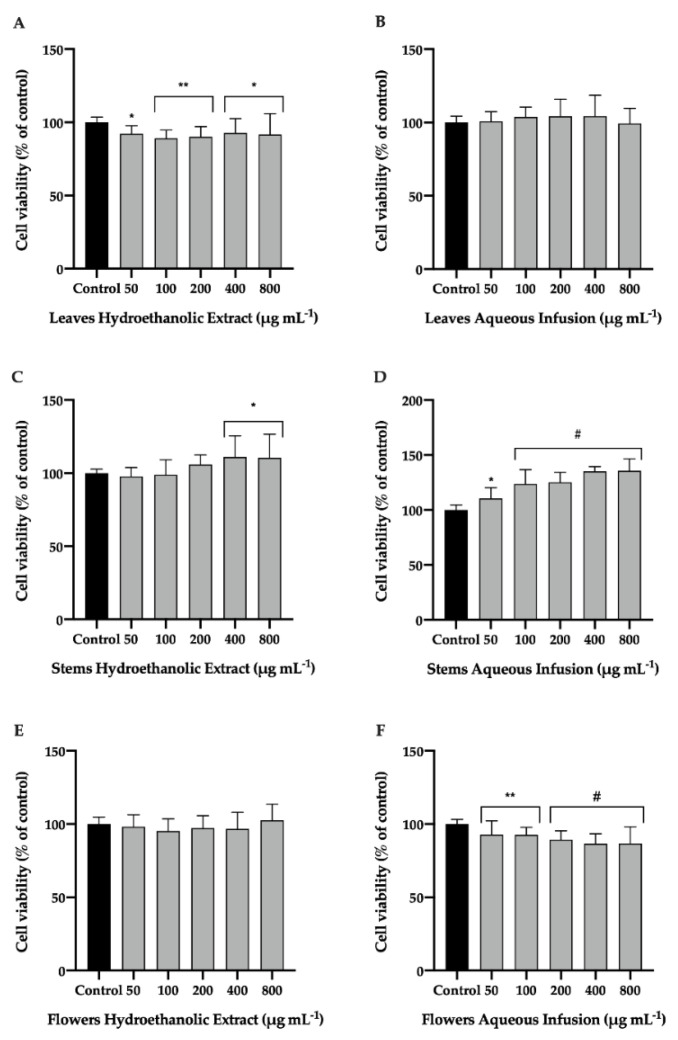
Effect of hydroethanolic extract (**A**,**C**,**E**) and aqueous infusion (**B**,**D**,**F**) of *Prunus avium* L. leaves, stems, and flowers on the normal human dermal fibroblast (NHDF) cell line. Cells were plated and exposed to increasing concentrations of extracts for 24 h. 3-(4,5-dimethylthiazol-2-yl)-2,5-diphenyltetrazolium bromide (MTT) assay was performed to assess cell viability. The presented data correspond to the means ± standard deviation of three independent assays and are represented as percentage (%) of control cells. Statistical analysis: One-way ANOVA was performed for each concentration compared to control; * *p* value < 0.05, ** *p* < 0.01, and # *p* < 0.001 were considered significant.

**Table 1 nutrients-14-04576-t001:** Diameters (mm) of inhibition zones for disk-diffusion method of hydroethanolic extracts and aqueous infusions from *P. avium* by-products.

	Hydroethanolic Extracts			Aqueous Infusions			Control
	Leaves	Stems	Flowers	Leaves	Stems	Flowers	Gentamycin
**Gram-positive**							
*M. luteus* CECT 243	0.00 ± 0.00	1.86 ± 0.25	1.00 ± 0.08	0.96 ± 0.05	1.16 ± 0.19	0.00 ± 0.00	15.83 ± 0.17
*E. faecalis* ATCC 29212	0.00 ± 0.00	2.85 ± 0.23	0.00 ± 0.00	0.00 ± 0.00	0.80 ± 0.21	0.00 ± 0.00	11.05 ± 0.38
*B. cereus* ATCC 11778	1.37 ± 0.17	7.24 ± 0.14	1.33 ± 0.17	1.14 ± 0.03	4.06 ± 0.2	1.20 ± 0.05	15.60 ± 0.82
*L. monocytogenes* LMG 16779	0.00 ± 0.00	1.94 ± 0.19	0.00 ± 0.00	0.00 ± 0.00	0.99 ± 0.01	0.00 ± 0.00	15.93 ± 0.13
*S. aureus* ATCC 25923	0.00 ± 0.00	3.04 ± 0.16	0.00 ± 0.00	0.00 ± 0.00	0.86 ± 0.09	0.00 ± 0.00	12.93 ± 0.43
*B. subtilis* CECT 35	0.00 ± 0.00	0.95 ± 0.06	0.76 ± 0.13	0.00 ± 0.00	0.64 ± 0.16	0.00 ± 0.00	29.70 ± 0.88
**Gram-negative**							
*S. tiphymurium* ATCC 13311	0.00 ± 0.00	0.00 ± 0.00	0.00 ± 0.00	0.00 ± 0.00	0.00 ± 0.00	0.00 ± 0.00	13.30 ± 0.65
*P. aeruginosa* ATCC 27853	0.00 ± 0.00	0.00 ± 0.00	0.00 ± 0.00	0.00 ± 0.00	0.00 ± 0.00	0.00 ± 0.00	18.03 ± 0.12
*E. coli* ATCC 25922	0.00 ± 0.00	0.00 ± 0.00	0.00 ± 0.00	0.00 ± 0.00	0.00 ± 0.00	0.00 ± 0.00	12.57 ± 0.22
*K. pneumoniae* ATCC 13883	0.00 ± 0.00	0.00 ± 0.00	0.00 ± 0.00	0.00 ± 0.00	0.00 ± 0.00	0.00 ± 0.00	10.30 ± 0.08
*P. mirabilis* CECT 17	0.32 ± 0.15	0.00 ± 0.00	0.00 ± 0.00	0.00 ± 0.00	0.00 ± 0.00	0.00 ± 0.00	17.73 ± 0.18
*S. marcescens* CECT 159	0.00 ± 0.00	0.00 ± 0.00	0.00 ± 0.00	0.00 ± 0.00	0.00 ± 0.00	0.00 ± 0.00	18.83 ± 0.09
*A. baumannii* LMG 1025	0.00 ± 0.00	0.00 ± 0.00	0.00 ± 0.00	0.00 ± 0.00	0.00 ± 0.00	0.00 ± 0.00	19.33 ± 0.27

Values for inhibition zone are presented as means ± standard deviations; ATCC—American Type Culture Collection; LMG—Laboratory for Microbiology of the Faculty of Sciences of the Ghent University; CECT—Spanish Type Culture Collection.

**Table 2 nutrients-14-04576-t002:** Minimum inhibitory concentration (MIC) values (mg mL^−1^) of hydroethanolic extracts and aqueous infusions from *Prunus avium* L. by-products.

	Hydroethanolic Extracts			Aqueous Infusions			Negative Control	Positive Control
	Leaves	Stems	Flowers	Leaves	Stems	Flowers	DMSO	Gentamicin
**Gram-positive**								
*M. luteus* CECT 243	0.5	2	2	2	1	>2	>2	0.016
*E. faecalis* ATCC 29212	0.016	0.5	2	0.016	1	>2	>2	0.016
*B. cereus* ATCC 11778	0.016	1	2	0.016	1	2	>2	0.016
*L. monocytogenes* LMG 16779	0.016	1	0.031	0.5	0.5	>2	>2	0.016
*S. aureus* ATCC 25923	0.5	2	2	2	2	2	>2	0.016
*B. subtilis* CECT 35	1	2	>2	2	2	>2	>2	0.016
**Gram-negative**								
*S. tiphymurium* ATCC 13311	0.016	0.031	2	0.016	0.031	1	>2	0.016
*P. aeruginosa* ATCC 27853	1	0.5	0.5	2	0.5	2	>2	0.016
*E. coli* ATCC 25922	0.5	1	1	2	1	2	>2	0.016
*K. pneumoniae* ATCC 13883	1	1	2	0.016	1	>2	>2	0.016
*P. mirabilis* CECT 17	0.016	2	>2	1	1	>2	>2	0.016
*S. marcescens* CECT 159	0.016	1	2	0.016	1	>2	>2	0.016
*A. baumannii* LMG 1025	0.016	1	>2	0.25	1	2	>2	0.016

## Data Availability

Not applicable.

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
