# Peer review of "Anti-Inflammatory and Antimicrobial Activities of Portuguese Prunus avium L. (Sweet Cherry) By-Products Extracts"

_nutrients, 2022, doi:10.3390/nu14214576_

Round 1
Reviewer 1 Report
The manuscript entitled “Anti-Inflammatory and Antimicrobial Activities of Portuguese 2 Prunus avium L. (Sweet Cherry) By-products Extracts” by Nunes et al. is concerned with biological evaluation of leaves, stems, and flowers of P. avium from the Fundão region (Portugal), as anti-inflammatory and anti-bacterial agents.
The Major points are:-
1. Toxicity study should be provided.
2. Anti-bacterial activity should be compared with different potent standard FDA approved drugs as Ciprofloxacin.

Author Response
The authors would like to thank the Reviewers for carefully reviewing our manuscript and providing us with their comments and suggestions to improve the quality of the work. The following responses have been prepared to address all the reviewer comments in a point-by-point table template. Additionally, all modifications made in the manuscript addressing the comments of Reviewers are highlighted in yellow.
Reviewer |
Comment |
Response |
1 |
1. Toxicity study should be provided. |
The authors understand the reviewer’s comment. Nonetheless, in this work, we evaluate the cellular toxicity in mouse macrophage cells (RAW 264.7) and normal human dermal fibroblasts (NHDF) cell lines through MTT assay. This assay is commonly used in studies of cell toxicity, despite all its limitations. The obtained results, in the mentioned cell lines, showed that Prunus avium L. leaves, stems, and flowers aqueous infusion and hydroethanolic extract were not toxic in the concentration range used. In the previous published study (https://doi.org/10.3390/foods10061185), we demonstrated that the same extracts of cherry by-products present toxicity at concentrations above 200 μg mL-1 in the human epithelial colorectal adenocarcinoma (Caco-2) cells, inhibiting their growth. |
|
2. Anti-bacterial activity should be compared with different potent standard FDA approved drugs as Ciprofloxacin. |
The authors thank the reviewer’s comment that contribute to increase the quality of the work. It is certain that, when we evaluate the antimicrobial activity, we should use several antimicrobial agents against used microbial strains. In fact, in drug discovery and drug development steps, the proposed intervention should demonstrate better efficacy and fewer adverse effects compared with the standard medications to consider as a new treatment. In this work, we must indicate that the choice of gentamicin was taken into account because it is a broad-spectrum aminoglycoside antibiotic used to treat both Gram-positive and Gram-negative bacteria such as those used in this study. This antibiotic has been used as a positive control in other studies: - https://doi.org/10.5530/pj.2018.3.83; - https://doi.org/10.1016/j.heliyon.2019.e02126; - https://doi.org/10.4028/www.scientific.net/JBBBE.52.11; - https://doi.org/10.1016/j.micpath.2018.08.026; - https://doi.org/10.26656/fr.2017.4(2).303). |
Reviewer 2 Report
This work is interesting for by-product extracts application on a health perspective. Nevertheless, some similar sentence was found by the plagiarism checking. The checking file was an attachment file herewith. Please check it out. Also, some point needs to make a clarify by the authors following:
1. Leaves, stems, and flowers of P. avium are the seasonal plant in Portugal? why it could harvest only in April and June?
2. Based on the cell viability test, it seems only the stem composition of the plant was working when compared to the control. So, the authors need to discuss the different compositions of the cell toxic part too in the discussion section.
3. What are the significant methods between Hydroethanolic and Aqueous infusion on plant extraction? which one is the most effective for the plant by-product extraction?
Author Response
The authors would like to thank the Reviewers for carefully reviewing our manuscript and providing us with their comments and suggestions to improve the quality of the work. The following responses have been prepared to address all the reviewer comments in a point-by-point table template. Additionally, all modifications made in the manuscript addressing the comments of Reviewers are highlighted in yellow.
2 |
This work is interesting for by-product extracts application on a health perspective. Nevertheless, some similar sentence was found by the plagiarism checking. The checking file was an attachment file herewith. Please check it out. Also, some point needs to make a clarify by the authors following: |
The authors are grateful for the reviewer’s comment and suggestion. As suggested, the plagiarism was checked. (Please see now version of the revised manuscript). |
|
1. Leaves, stems, and flowers of P. avium are the seasonal plant in Portugal? why it could harvest only in April and June? |
The authors thank the reviewer’s comment. The Prunus avium L. belongs to the Rosacea family and its cultivation is widespread throughout almost the entire world, particularly in the Mediterranean area, namely in Portugal country. Sweet cherries are popular spring-summer fruits. In Portugal, namely in Fundão region, the cherry flowers appear usually in April and fall before the appearance of the fruit. Thus, it is during this month that flowers are harvested, after their fall. Furthermore, it is during the months of May and June that cherry fruit has the high point of production, so its leaves and stems were collected in this time. Further studies are being carried out to demonstrate possible differences in phytochemical composition and biological properties in, for example, leaves harvested at other times of the year. |
|
2. Based on the cell viability test, it seems only the stem composition of the plant was working when compared to the control. So, the authors need to discuss the different compositions of the cell toxic part too in the discussion section. |
The authors thank the reviewer’s suggestion. As suggested, the results and discussion about the stems composition and their impact in cell viability were revised and completed, contributing to increase the quality of work. (Please see now lines 234 to 245 of the revised manuscript). |
|
3. What are the significant methods between Hydroethanolic and Aqueous infusion on plant extraction? which one is the most effective for the plant by-product extraction? |
The authors thank the reviewer’s comment. In recent years, there has been a growing interest in the bioavailability and biological effects of different phytochemicals existents in plants, namely phenolic compounds. Leaves and stems of Prunus avium L. possess high phenolic content and have been used, since ancient times, in infusions and decoctions in traditional medicine. In this work, we used an aqueous infusion to simulate the herbal preparations in the daily life of people. In contrast, the hydroethanolic extract was chosen because it allows better extraction of phenolic compounds, according to several reports in the literature. Water, methanol, hexane, acetone and ethanol are the most frequent solvents for extracting polyphenols. However, extraction with ethanol, especially at concentrations greater than 50%, is the most frequent extraction solvent, revealing to be particularly effective to obtain extracts rich in phenolics. In addition, most phytochemicals present in plants are molecules with phenolic structures, which are more easily extracted by polar mixtures, such as alcoholic solvents. |
Reviewer 3 Report
General comments:
The article investigated the anti-inflammatory and antimicrobial potential of two different extracts from stems, leaves, and flowers of Portuguese cherries. The anti-inflammatory activity was investigated on lipopolysaccharide (LPS)-stimulated mouse macrophages (RAW 264.7) by evaluating their effect on cellular viability and nitric oxide (NO) production. Disc diffusion and minimum inhibitory concentration (MIC) were used to determine antimicrobial activity. The structure of the paper is relatively complete and innovative. Some interesting results were provided. But there are some corrections required to improve the current form of the manuscript for publication.
1. The author should add the identification results of the active components of different extracts.
2. Line 101-102: “Samples were frozen, freeze-dried”. Please add relevant equipment and equipment information.
3. Line 104: Please describe the preparation process of the extract in detail.
4. Many units in the manuscript are not uniform, such as 24 h, 24 hours, etc. please revise the full text.
5. “H3PO4”?
6. Line 183: “incubated at 37 ◦C for 24 hours.” ? Line 192: 5 ×106?
7. Please put “3.1. Anti-inflammatory Activity ” into “3.1.1. Effect of P. avium by-products on the viability of the Raw 264.7 cell line ” for analysis.
8. “Resuls and Discussion”, in-depth discussion is needed and some sections need further clarification. Moreover, please provide corresponding literature support when explaining the reasons behind the experimental results. Please check the full text.

Author Response
The authors would like to thank the Reviewers for carefully reviewing our manuscript and providing us with their comments and suggestions to improve the quality of the work. The following responses have been prepared to address all the reviewer comments in a point-by-point table template. Additionally, all modifications made in the manuscript addressing the comments of Reviewers are highlighted in yellow.
3 |
The article investigated the anti-inflammatory and antimicrobial potential of two different extracts from stems, leaves, and flowers of Portuguese cherries. The anti-inflammatory activity was investigated on lipopolysaccharide (LPS)-stimulated mouse macrophages (RAW 264.7) by evaluating their effect on cellular viability and nitric oxide (NO) production. Disc diffusion and minimum inhibitory concentration (MIC) were used to determine antimicrobial activity. The structure of the paper is relatively complete and innovative. Some interesting results were provided. But there are some corrections required to improve the current form of the manuscript for publication. |
The authors sincerely thank the Reviewer for the overall appreciation of the work and for the comments and suggestions that contributed to increase the quality of the manuscript. |
|
1. The author should add the identification results of the active components of different extracts. |
The authors thank the reviewer’s suggested. The phytochemical composition of different extracts was previously studied and the results were published in the following articles: - https://doi.org/10.3390/foods10061185; - https://doi:10.3390/foods11050751. However, the most important bioactive compounds of different extracts have been added and discussed in the results section (Please see now the Results and Discussion section of revised version of manuscript). |
|
2. Line 101-102: “Samples were frozen, freeze-dried”. Please add relevant equipment and equipment information. |
The reference to the equipment used is described in the manuscript. (Please see now lines 101 and 102 of the revised manuscript). |
|
3. Line 104: Please describe the preparation process of the extract in detail. |
As suggested, the preparation process of the extract was revised and described in detail. (Please see now lines 104 to 117 of the revised manuscript). |
|
4. Many units in the manuscript are not uniform, such as 24 h, 24 hours, etc. please revise the full text |
The manuscript was carefully revised and the units were uniformized. |
|
5. “H3PO4”? |
The chemical formula of phosphoric acid was corrected. |
|
6. Line 183: “incubated at 37 ◦C for 24 hours.” ? Line 192: 5 ×106? |
The line 183 was revised. (Please see now lines 186 to 188 of the revised manuscript). Regarding the line 192, the authors do not understand what the reviewer intends because the description “5 x 106” does not exist in the manuscript. Please consider the revised version of the manuscript. Thank you. |
|
7. Please put “3.1. Anti-inflammatory Activity ” into “3.1.1. Effect of P. avium by-products on the viability of the Raw 264.7 cell line ” for analysis. |
The authors thank the reviewer’s suggestion. However, the section “3.1. Anti-inflammatory Potential” is a brief introduction to the following subsections regarding anti-inflammatory activity. Therefore, we ask that you consider this approach correct. Thank you. |
|
8. “Results and Discussion”, in-depth discussion is needed and some sections need further clarification. Moreover, please provide corresponding literature support when explaining the reasons behind the experimental results. Please check the full text. |
The section “Results and Discussion” was revised and completed in order to increase the clarity and interpretation of experimental results based on the reviewer’s comment, and corresponding literature support was included. (Please see now the Results and Discussion section of revised version of manuscript). |
Round 2
Reviewer 3 Report
The author carefully revised the manuscript. I think the current manuscript can be accepted.